# Investigating Lifted Heuristics for Timeline-based Planning

**Riccardo De Benedictis** and **Amedeo Cesta**

CNR - Italian National Research Council, Institute of Cognitive Sciences and Technologies
Via San Martino della Battaglia 44, 00185, Rome, Italy
{name.surname}@istc.cnr.it

## Abstract

This paper investigates the use of lifted heuristics, inspired by the more classical ones for the resolution of STRIPS-like problems, for the efficient resolution of *timeline-based* planning problems. We propose, in particular, a new heuristic strategy which, while maintaining the variables lifted, allows more accurate decisions. Furthermore, the concepts presented in this work pave the way for a new type of heuristics which, at present, allow this kind of solvers a significant performance improvement.

## Introduction

Since their early introduction, domain-independent heuristics have immediately proven to be a fundamental ally in solving difficult combinatorial problems such as those related to automated planning. The number of heuristics, introduced in recent years, for the efficient resolution of these problems has grown significantly to the point of constituting a research field (called *heuristic planning*) in its own. The different approaches that make up a solver's paraphernalia, range from the seminal $h_{add}$ and $h_{max}$ (Bonet and Geffner 2001) to the more recent developments relying on *delete-relaxation*, like the $h^{FF}$ heuristic (Hoffmann and Nebel 2001) and the *causal graph* heuristics (Helmert 2006), on *landmarks*, like in (Hoffmann, Porteous, and Sebastia 2004; Porteous, Sebastia, and Hoffmann 2014), on the *critical path*, like the $h^m$ heuristic (Haslum and Geffner 2000; Haslum, Bonet, and Geffner 2005) or, lastly, on *abstraction*, like in (Edelkamp 2014) or in (Helmert, Haslum, and Hoffmann 2007; Helmert et al. 2014).

While the above heuristics are significantly heterogeneous among them (although, often, they share some commonalities), they have in common the fact that they have been developed specifically for the resolution of a particular type of problem, characterized by a specific modeling language called PDDL (Ghallab et al. 1998), representing a natural evolution of the most long-lived STRIPS (Fikes and Nilsson 1971) formalism. Despite the PDDL, over the years, has been extended through different directions by introducing *durative-actions* and *numeric fluents* (Fox and Long 2003), *derived predicates* and *timed initial literals* (Edelkamp and Hoffmann 2004), *continuous changes* (Fox and Long 2006), *state-trajectory constraints* and *preferences* (Gerevini et al. 2009) and *object-fluents*[1], the development of heuristics for reasoning with these more expressive formal systems has remained relatively limited to a few cases (e.g., (Piotrowski et al. 2016; Franco et al. 2019)).

Although it significantly departs from the previous ones, the *timeline-based* approach represents a different formalism that, already in its original formulation (Muscettola et al. 1992), is able to cover a large part of the above features. Although introduced before the aforementioned formalisms, this specific planning paradigm has always remained a niche within the automated planning community. The fragmentation of the different timeline-based formalisms, indeed, did not allow the emergence of a common language which would have enabled a fair comparison among the different reasoners. Furthermore, analogously to the solvers reasoning upon the previous PDDL extensions, timeline-based planners have to cope with the high expressiveness of the formalisms which, despite making them particularly suited at addressing real-world applications, unavoidably leads to performance issues. The contribution of this paper, a slightly modified version of (De Benedictis and Cesta 2020), is, hence, twofold: after providing a new formalization of the timeline-based problem, aiming to embracing the different aspects of the previous formalisms, we propose a new domain-independent heuristic which, inspired by the more classical ones, aims at improving the resolution efficiency.

## Timeline-based planning

Timeline-based planning was first introduced in (Muscettola et al. 1992; Muscettola 1994) and, since then, many solvers have been proposed like, for example, ᴋTᴇT (Ghallab and Laruelle 1994), Eᴜʀᴏᴘᴀ (Jonsson et al. 2000), Aꜱᴘᴇɴ (Chien et al. 2010), the Tʀꜰ (Fratini, Pecora, and Cesta 2008; Cesta et al. 2009) on which the APSI framework (Fratini et al. 2011) relies and, more recently, PLAT-INUm (Umbrico et al. 2017). Some theoretical work on timeline-based planning like (Frank and Jónsson 2003; Jonsson et al. 2000) was mostly dedicated to identifying connections with classical planning a-la PDDL (Fox and Long

---

[1] http://www.plg.inf.uc3m.es/ipc2011-deterministic/attachments/Resources/kovacs-pddl-3.1-2011.pdf

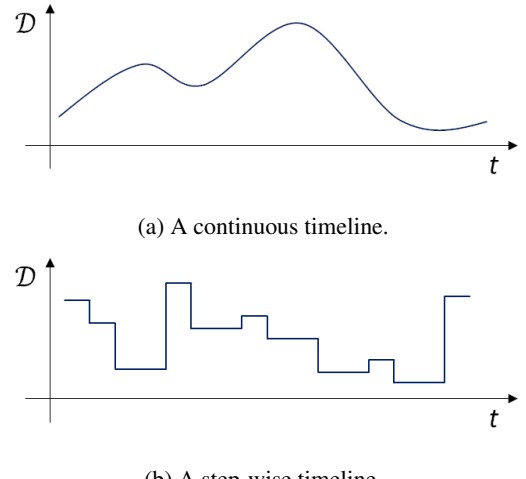

(a) A continuous timeline.

(b) A step-wise timeline.

Figure 1: A continuous and a step-wise timeline.

2003). A recent new formalization of timeline-based planning has been proposed in (Cialdea Mayer, Orlandini, and Umbrico 2016), while (Gigante et al. 2020) studied its properties from a computational complexity point of view. The work on ᴋᴛᴇᴛ and Tʀꜰ has tried to clarify some key underlying principles but mostly succeeded in underscoring the role of time and resource reasoning (Cesta and Oddi 1996; Laborie 2003). The planner CHIMP (Stock et al. 2015) follows a Meta-CSP approach having meta-Constraints which havely resembles timelines. The Flexible Acting and Planning Environment (FAPE) (Dvořák et al. 2014) tightly integrates timelines with acting. The Action Notation Modeling Language (ANML) (Smith, Frank, and Cushing 2008) is an interesting development which combines the HTN decomposition methods with the expressiveness of the timeline representation. Finally, it is worth mentioning that the timeline-based approaches have been often associated to resource managing capabilities. By leveraging on constraint-based approaches, most of the above approaches like ᴋᴛᴇᴛ (Laborie and Ghallab 1995; Laborie 2003), (Cesta, Oddi, and Smith 2002), (Smith, Frank, and Jónsson 2000) or (Verfaillie, Pralet, and Lemaître 2010) integrate planning and scheduling capabilities.

In order to better understand what we are talking about when discussing about timeline-based planning, it is important to introduce, without going into too much formal details, some basic concepts about *constraint networks* (Dechter 2003; Lecoutre 2009). Some of the timeline-based frameworks like, for example, those described in (Smith, Frank, and Jónsson 2000; Frank and Jónsson 2003), refer to timeline-based planning in terms of constraint-based planning, further emphasizing the central role that constraints take on within this type of planning. Formally,

**Definition 1.** *A constraint network $\mathcal{N}$ is composed of a finite set of variables, denoted by $vars(\mathcal{N})$, and a finite set of constraints, denoted by $cons(\mathcal{N})$.*

Specifically, constraint networks represent the lowest level elements on which timeline-based planning relies. The main data structure for the timeline-based paradigm is, indeed, the *timeline* which, in generic terms, is a function of time, either discrete or continuous, over a given domain. Formally,

**Definition 2.** *A timeline $\mathsf{T}$ is a function*

$$\mathsf{T} : \mathbb{T} \to \mathcal{D}$$

*where $\mathbb{T}$ is the (either discrete or continuous) domain of time and $\mathcal{D}$ is the (possibly infinite) domain of the timeline.*

It is worth noticing that the previous definition is quite general, not specifying any limitation neither on the time, which can be either discrete or continuous, nor on the domain which can be, in general, of any kind. Specifically, the domain of a timeline can be either symbolic (e.g., "a", "b", "c", etc.) or numeric (e.g., "1", "2", "3", etc.). Additionally, numeric domains can be either integer (e.g., "10", "12", "25", etc.) or real (e.g., "1.23", "2.17", "3.14", etc.). While integer domains can change in time only step-wise, real domains can change both step-wise and continuously. Finally, continuous changes can happen both linearly or non-linearly. Figure 1 (a), for example, represents a continuously updating non-linear timeline over reals. Figure 1 (b), on the contrary, shows a step-wise updating timeline.

Since the definition of timeline is completely general, it is possible to represent, through these, extremely heterogeneous concepts. We need, therefore, a unifying element that allows to represent contents homogeneously, in a way which is agnostic from the nature of the timeline. To this end, we introduce the concept of *token* and establish that values on timelines are a direct consequence of tokens through a timeline extraction procedure (more details soon). Without loss of generality, a token is an "assertion over a temporal interval". Formally,

**Definition 3.** *A token is an expression of the form:*

$$n(x_0, \ldots, x_i)_\chi @ [s, e, \tau]$$

*where $n$ is a predicate name, $x_0, \ldots, x_i$ are the parameters of the predicate (i.e., constants, numeric variables or symbolic variables), $\chi$ is the class of the token (i.e., either a fact or a goal), $s$ and $e$ are the temporal parameters of the token (i.e., constants or variables) belonging to $\mathbb{T}$ such that $s \leq e$ and $\tau$ is the scope parameter of the token (i.e., a constant or a symbolic variable) representing the timeline on which the token apply.*

Roughly speaking, the expression on the left of the "@" symbol represents the "assertion" while the expression at its right represents the "interval". In other words, a token $n(x_0, \ldots, x_i)_\chi @ [s, e, \tau]$ asserts that $\forall t$ such that $s \leq t \leq e$, the relation $n(x_0, \ldots, x_i)$ holds at the time $t$ on the timeline $\tau$. Furthermore, given a token $\eta$, we call $pars(\eta)$ its parameters $x_0, \ldots, x_i, s, e, \tau$.

Tokens constitute the main building blocks of timeline-based plans. Regardless of the resolution procedure, indeed, the role of any timeline-based solver consists in introducing new tokens and/or establishing the values of their parameters. A critical aspect to keep in mind, when talking about

tokens, is that, in general, their parameters are variables of a constraint network and, as such, can be constrained. In other words, in order to reduce the allowed values for the tokens' constituting parameters, and thus decreasing the modeled system's allowed behaviors, it is possible to impose *constraints* among them (and/or among the parameters and other possible variables). Such constraints include temporal constraints, binding constraints between symbolic variables as well as (non)linear constraints among numerical variables (possibly including temporal variables).

The set of tokens and constraints is used to describe the main data structure that is used to represent (partial) plans of the timeline-based approach: the *token network*. Formally,

**Definition 4.** *A token network is a tuple* $\pi = (\mathcal{T}, \mathcal{N})$, *where:*

- $\mathcal{T} = \{\eta_0, \ldots, \eta_j\}$ *is a set of tokens, such that* $\forall \eta \in \mathcal{T}, pars(\eta) \subseteq vars(\mathcal{N})$.
- $\mathcal{N}$ *is a constraint network.*

Finally, as already mentioned, tokens can be partitioned into two classes: *facts* and *goals*. While facts are, by definition, inherently true, goals have to be achieved. Causality, in particular, in the timeline-based approach, is defined by means of a set o *rules* indicating how to achieve goals. Formally,

**Definition 5.** *A* rule *is an expression of the form*

$$n(x_0, \ldots, x_k) @ [s, e, \tau] \leftarrow r$$

*where:*

- $n(x_0, \ldots, x_k) @ [s, e, \tau]$ *is the* head *of the rule, i.e. an expression in which* $n$ *is a predicate name,* $x_0, \ldots, x_k$ *are the parameters of the head (i.e., numeric variables or symbolic variables),* $s$ *and* $e$ *are the* temporal parameters *of the head (i.e., constants or variables) belonging to* $\mathbb{T}$ *such that* $s \leq e$ *and* $\tau$ *is the* scope parameter *of the head (i.e., a constant or a symbolic variable) representing the timeline on which the rule apply.*
- $r$ *is the* body *of the rule (or the* requirement*), i.e. either another token, a* constraint *among tokens (possibly including the* $x_0, \ldots, x_k, s, e, \tau$ *variables), a* conjunction *of requirements or a (priced[2]) disjunction[3] of requirements.*

Specifically, rules define causal relations that must be complied to in order for a given goal to be achieved. Roughly speaking, for each goal having the "form" of the head of a rule, the body of the rule (i.e., a logic combination of further tokens and constraints) must also be present in the token network. An example of rule is given by

---

[2]It is possible, if needed, to associate a cost to the different disjuncts of a disjunction so as to model preferences.

[3]Some formalisms allow the definition of different rules having the same head, thus modeling the disjunctions. We preferred to replace this possibility by explicitly representing disjunctions. This choice can, in cases where these rules share some of the requirements, favor the modeler by reducing the size of the domain.

$$At(?x) @ [s, e, \tau] \leftarrow \left\{ \begin{array}{c} [e - s \geq 1] \wedge \\ \left\{ \begin{array}{c} \left\{ \begin{array}{c} dt : DriveTo(?x)_g @ [s, e, \tau] \wedge \\ [\tau == dt.\tau] \wedge [s == dt.e] \wedge \\ [?x == dt.?x] \end{array} \right\} \vee \\ \left\{ \begin{array}{c} ft : FlyTo(?x)_g @ [s, e, \tau] \wedge \\ [\tau == ft.\tau] \wedge [s == ft.e] \wedge \\ [?x == ft.?x] \end{array} \right\} \end{array} \right\} \end{array} \right\}$$

By combining tokens, constraints, conjunctions and disjunctions, the above rule states that, in order to be in a given position, our agent must reach it either by driving or by flying.

We have now all the ingredients to define a timeline-based planning problem. In particular, the definition can rely on the above concept of requirement.

**Definition 6.** *A* timeline-based *planning problem is a triple* $\mathcal{P} = (\mathbf{T}, \mathcal{R}, r)$, *where:*

- $\mathbf{T}$ *is a set of timelines.*
- $\mathcal{R}$ *is a set of rules.*
- $r$ *is a requirement, i.e. either a (fact or goal) token, a constraint among tokens, a conjunction of requirements or a (priced) disjunction of requirements.*

It is worth highlighting that, conversely to other timeline-based approaches, our formalism makes a clear distinction between tokens and values on timelines. This difference aims at guaranteeing us a further element of generality. The transition from tokens to timelines, however, requires the introduction of a further function which allows to *extract* the timelines from the tokens. Specifically,

**Definition 7.** *An* extraction function $\mathsf{X}_\mathsf{T}$ *is a function for a timeline* $\mathsf{T}$

$$\mathsf{X}_\mathsf{T} : \mathbb{T} \times 2^{\mathcal{T}_\mathsf{T}} \to \mathcal{D}$$

*where* $\mathbb{T}$ *is the (either discrete or continuous) domain of time,* $\mathcal{T}_\mathsf{T}$ *is the set of tokens in the token network, having* $\mathsf{T}$ *in the domain of their* $\tau$ *variable, and* $\mathcal{D}$ *is the domain of the timeline.*

As can be easily seen by comparing Definition 2 with Definition 7, the result of the extraction function is, basically, a timeline. Each type of timeline, indeed, has associated its own timeline extraction procedure which allows to pass from the associated tokens to the resulting timelines. In other words, the timeline extraction procedure assigns to the tokens a higher-level semantic: according to the nature of the timeline, the procedure is able to "recognize the meaning" of the involved tokens. Note that, thanks to the introduction of the above higher-level semantic, not all token configurations lead to consistent timelines. According to the nature of the timeline, indeed, some configurations of tokens might lead to inconsistencies. It is responsibility of the solver to introduce further constraints so as to avoid such inconsistencies. Another way to see a timeline, indeed, is in terms of a *global constraint* (refer, for example, to (Dechter 2003; Lecoutre 2009)) over those tokens of the token network which assume the same value for their $\tau$ variables. Such global constraints, in particular, depend on the nature of the timeline, hence justifying the introduction of this concept within the formalism.

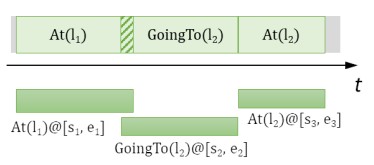

(a) A state-variable timeline.

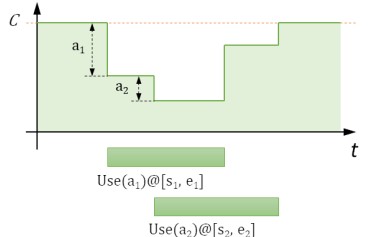

(b) A reusable-resource timeline.

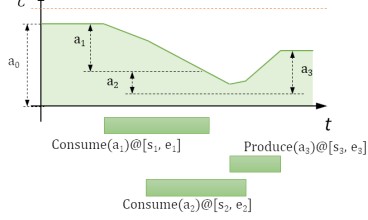

(c) A consumable-resource timeline.

Figure 2: Different timelines extracted by tokens.

Examples of timelines, extracted from tokens, are shown in the Figure 2. Specifically, Figure 2a shows a *state-variable* timeline, a step-wise timeline whose domain depends from the tokens which can be assigned, by means of the $\tau$ variable (omitted, for simplicity), to it. This type of timeline, in particular, introduces an additional global constraint that guarantees that different values, on the same timeline, cannot overlap in time. The state-variable of Figure 2a, as an example, has two values that overlap as a consequence of the overlapping of the $At(l_1)$ and the $GoingTo(l_2)$ tokens. Such an inconsistency can be solved, for example, by imposing an ordering constraint between the tokens (e.g., $e_1 \leq s_2$). Another type of timeline, typically used in pure scheduling problems, is the *reusable-resource* (see Figure 2b). This step-wise timeline is characterized by a maximum capacity and by a resource level which changes over time according to how the tokens, representing resource usages, overlap. The resource constraint guarantees that concurrent uses of the resource do not exceed its capacity. Finally, as example of a continuous timeline, the *consumable-resource* timeline (see Figure 2c) is characterized by a maximum capacity and by an initial amount. Similarly to reusable-resources, the resource level changes over time according to how the tokens, representing resource productions and consumptions, overlap, while the resource constraint guarantees that the level never exceeds the resource capacity nor goes below zero.

It is worth noticing that, unlike existing formalizations, by enabling any implementing solver to reason about timelines agnostically from their specific nature, the above definition allows us to maintain a certain generality. Furthermore, once provided an extraction function and the algorithms for managing the specific global constraint, new types of timelines can be introduced without affecting the solvers' resolution procedures.

The last aspect to consider regards the solution of a timeline-based planning problem. Roughly speaking, a solution is a token network whose all goals have been achieved. Furthermore, at least one *consistent* (i.e., does not violate any constraint) and *complete* (i.e., it includes all the variables) assignment of values to the variables of the underlying constraint network must be available. Notice that, among the constraints of the constraint network, there are also those which are imposed by the timelines. Formally,

**Definition 8.** *A token network* $\pi = (\mathcal{T}, \mathcal{N})$ *is a* solution *for a timeline-based planning problem* $\mathcal{P} = (\mathbf{T}, \mathcal{R}, \boldsymbol{r})$ *if:*

– *there exists a complete and consistent assignment of values to the variables of the constraint network* $\mathcal{N}$.
– *every goal* $g \in \mathcal{T}$ *is achieved (i.e., either the goal* $g$ *is recognized as semantically equivalent to another token, or a rule, whose head is compatible with the token* $g$, *is applied).*

## Reasoning with timelines

Unfortunately, the above definitions do not provide a computable test for building and verifying solutions. This section, therefore, introduces the typical approach for solving timeline-based planning problems. Specifically, common timeline-based solvers strongly rely on partial-order planning (Weld 1994) for reasoning, generalizing the concept of *threat* for including any possible inconsistency which might arise as a consequence of the timeline constraints (e.g., different states overlapping on the same state-variable, resources overusages, etc.). Despite this generalization, the search space (and, consequently, the solving algorithm) remains substantially unchanged. In particular, timeline-based solvers rely on the concept of *flaws*, that a token network has, and on the concept of *resolvers*, for solving them. Formally,

**Definition 9.** *A* flaw *in a token network* $\pi = (\mathcal{T}, \mathcal{N})$ *is either: (i) an* open goal *(i.e., a goal whose associated rule has not yet been applied or which has not yet been recognized as semantically equivalent to another token), (ii) a* threat *(i.e., any possible inconsistency arising as a consequence of the timeline constraints) or (iii) a* disjunction.

Intuitively, the main resolution principle consists in refining the token network $\pi$, identifying its flaws and applying resolvers for solving them, while maintaining the constraints $cons(\mathcal{N})$ consistent, until the token network $\pi$ has no more flaws.

Figure 3 specifies a recursive non-deterministic procedure called TP (for Timeline-based Planning) for resolving timeline-based planning problems. Specifically:

– *flaws* denotes the set of all flaws in $\pi$ provided by procedures `OpenGoals`, `Threats` and `Disjunctions`; $\varphi$ is a particular flaw in this set.

```
procedure TP(π)
    flaws ← OpenGoals (π) ∪ Threats (π) ∪ Disjunctions (π)
    if flaws = ∅ then return π
    end if
    select any flaw φ ∈ flaws
    resolvers ← Resolve (φ, π)
    if resolvers = ∅ then return failure
    end if
    non-deterministically choose a resolver ρ ∈ resolvers
    π' ← Refine (ρ, π)
        return TP (π')
end procedure
```

Figure 3: The TP procedure for solving timeline-based planning problems.

– *resolvers* denotes the set of all possible ways to resolve a specific flaw $\varphi$ in a plan $\pi$ and is given by the procedure `Resolve`. The resolver $\rho$ is a particular element of this set.

– $\pi'$ is the new plan obtained by refining $\pi$ according to the resolver $\rho$ as a consequence of the procedure `Refine`.

The TP procedure is called with an initial token network $\pi_0$, characterized by the problem's requirement. Each successful recursion is a refinement of the current plan according to the chosen resolver. In particular, the `Resolve` procedure returns all the resolvers that, in the token network $\pi$, solve the $\varphi$ flaw. These resolvers depend, necessarily, on the type of flaw $\varphi$ and on the current token network $\pi$. In the case of open goals, for example, resolvers represent the application of the corresponding rule or the *unification* (i.e., same predicate name and same, pairwise, parameter values, hence recognizing the tokens as semantically equivalent) with another already achieved goal or fact. In the case, for example, of excessive concurrent resource usages, conversely, resolvers could represent ordering constraints between couples of tokens. As a consequence, each invocation of the `Refine` procedure might introduce new tokens, new variables and/or new constraints to the token network. Intuitively, refinement operations should be chosen so as to avoid adding to the token network any constraint that is not strictly needed (this is called the *least commitment principle*).

## Toward more effective heuristics

Reasoning within the above formal system is not at all simple[4]. It is worth noting that while the choice of the resolver is a *non-deterministic* step (i.e., it may be required to backtrack on this choice), the selection of a flaw is a *deterministic* step (i.e., there is no reason to backtrack on this choice) as all flaws need to be solved before or later in order to reach a solution plan. Despite the order in which flaws are processed is very important for the efficiency of the procedure, it is unimportant for its soundness and completeness. A deterministic implementation of the TP procedure should rely on algorithms like A* or IDA* so as to avoid that the search may

---

[4]Note that it is possible, in general, to represent through this formalism a self-referential proposition $P$, whose meaning is "$P$ is false", hence showing the formalism's undecidability.

keep exploring deeper and deeper a single path in the search space, adding indefinitely new tokens to the partial plan and never backtracking. As a consequence, choosing the *right* flaw and the *right* resolver becomes a crucial aspect for coping with the computational complexity and hence efficiently generating solutions.

The main difficulty derives from the impossibility of i) having a perfectly defined current state and ii) measuring the distance between this state and a desired state indicated in the formulation of the planning problem. For these reasons it becomes particularly inconvenient to use or even adapt, directly, the heuristics developed for classical formalisms. What we propose in this document is, somehow, to separate the temporal aspects from the purely causal ones, which in classical planning are strongly linked to be almost the same thing, and to apply classical heuristics only to the latter. In doing so, the rules of the timeline formalism become the equivalent of the PDDL operators, having the requirements as preconditions and the head of the rule as the only positive effect. Once this paradigm shift has been made, it becomes possible to adapt the heuristics of classical planning. Note that, however, this translation is not trivial: if, on the one hand, there is the simplification of having, for each operator, only a single positive effect (i.e., the solved flaw), on the other hand there is the difficulty of rendering atoms "ground" due to the presence of numerical parameters (representing, for example, the starting and the ending times of the tokens). We are therefore forced to reason about a sort of causal graph having lifted variables.

The overall proposed idea consists in applying, in a coarse way, *all* the possible resolvers for *all* the possible flaws until some termination criteria, i.e., unifications and resolvers which do not add further flaws, is met. Specifically, since flaws and resolvers are causally related (i.e., resolvers might introduce flaws which are solved by other resolvers, etc.) it is possible to build an AND/OR graph for representing such causal relations. By doing so, instead of searching in the space of the token networks, we have a single disjunctive token network containing all the possible plans (or, hopefully, only the "most interesting" ones) that can be found starting from the initial token network $\pi_0$. By exploiting the topology of such a graph it is possible to generate an estimation of "how far" a flaw is from being solved and exploit this estimation for guiding the resolution process. Specifically, taking inspiration from the $h_{add}$ and the $h_{max}$ heuristics introduced in (Bonet and Geffner 2001), the cost of a resolver, which can be seen as an AND node, can be estimated as the *maximum* (in case of $h_{max}$ heuristic, or the *sum*, in case of the $h_{add}$ heuristic) of the estimated costs of the flaws introduced by the resolver itself plus an intrinsic resolver's cost, while the estimated cost of a flaw, which can be seen as an OR node, can be estimated as the *minimum* of the estimated costs of its possible resolvers. Since all flaws must be solved, the solver chooses, among those that have to yet been solved, the most expensive one (i.e., the one that, most likely, will detect an inconsistency earlier) and will solve it with the least expensive resolver (i.e., the one that, more likely, will lead to a solution).

# The lifted heuristic formulation

Before formally introducing the proposed heuristics, it is worth providing some definitions. Specifically, since the presence of flaws and resolvers, within the current partial solution, is controlled by a set of propositional variables, we refer to flaws by means of $\varphi$ variables (we will use subscripts to describe specific flaws, e.g., $\varphi_0$, $\varphi_1$, etc.) and to resolvers by means of $\rho$ variables (similarly to flaws, we will use subscripts to describe specific resolvers, e.g., $\rho_0$, $\rho_1$, etc.). Specifically, the value of such variables will be used to recognize active flaws that have to be solved (i.e., those flaws whose $\varphi$ variables assume the $true$ value) and applied resolvers (i.e., those resolvers whose $\rho$ variables assume the $true$ value). Additionally, given a flaw $\varphi$, we refer to the set of its possible resolvers by means of $res(\varphi)$ and to the (possibly empty) set of resolvers which are responsible for introducing it by means of $cause(\varphi)$. The latter set is usually constituted by the sole resolver representing the application of the rule which introduced the flaw. Nonetheless, this set can also be empty in case of top-level flaws, in which case the $true$ value is assigned to their controlling $\varphi$ variables or, also, can contain more than one resolver in case the flaw is a consequence of their simultaneous application (e.g., a flaw representing two states overlapping on the same state-variable is activated whenever the rules that introduce the two states are applied simultaneously). Finally, given a resolver $\rho$, we refer to the set of its preconditions (e.g., the set of tokens introduced by the application of a rule) by means of $precs(\rho)$ and to the flaw solved through its application by means of $eff(\rho)$.

The above definitions allow us to formally introduce our heuristics. Specifically, let $G$ be the estimated cost function, the estimated cost of a flaw $\varphi$ and of a resolver $\rho$ are characterized by the following equations:

$$G(\varphi) = min_{\rho \in res(\varphi)} G(\rho)$$
$$G(\rho) = c(\rho) + max_{\varphi \in precs(\rho)} G(\varphi)$$

where $c(\rho)$ is the *intrinsic cost* of the $\rho$ resolver, i.e., a positive number representing the "cost" of disjuncts, in case of priced disjunctions, or the value 1, in other cases.

Similar to planning models based on satisfiability (Kautz and Selman 1992), it is possible to introduce propositional constraints to the $\varphi$ and $\rho$ variables so as to guarantee the causal relations. By doing so, once the graph has been built, it is possible to frame the search space within a given boundary, dropping the computational complexity of the search procedure to a "simpler" NP-hard[5]. Furthermore, the introduction of these variables allows the use of propagation techniques and, in the event of inconsistencies, conflict analysis (and, hence, non-chronological backtracking) techniques, typical of SAT/SMT based solvers. The planning problem is therefore reduced to the assignment of $true$ values to the

---

[5]There is, intuitively, no guarantee that the built graph contains a solution. Similarly to what happens in Graphplan (Blum and Furst 1997), indeed, it might be required the addition of a "layer" to the graph.

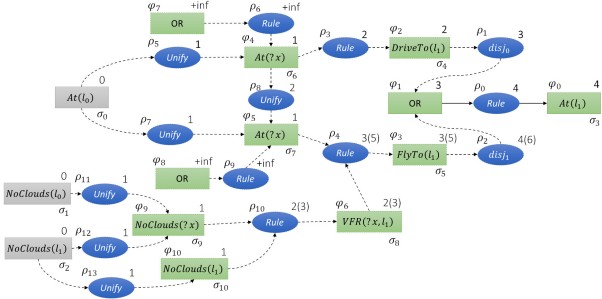

Figure 4: An example of causal graph with lifted variables.

variables associated to the resolvers while observing the assignment, as a consequence of constraint propagation, of $true$ values to the variables associated to the flaws.

Additionally, in order to establish the presence or not of the tokens inside the solution, a state variable $\sigma \in \{inactive, active, unified\}$ is associated to each token. A partial solution will hence consist solely of those tokens of the token network which are *active*. Moreover, in case such tokens are goals, the bodies of the associated rules must also be present within the solution. The *unified* tokens do not participate directly in the partial solution, since they are recognized as semantically equivalent to other active tokens, yet, since possibly subject to constraints, they might indirectly influence the "shape" of the solution. Finally, *inactive* tokens do not participate at all in the solution. We refer to tokens, later on, by means of $\sigma$ variables (we will use subscripts to describe specific tokens, e.g., $\sigma_0$, $\sigma_1$, etc.) and to the flaws introduced by tokens by means of the $\varphi(\sigma)$ function.

## An explanatory example

In order to better understand how the heuristic and causality constraints work we introduce, in this section, a very simple example involving an agent moving between different locations either by driving or by flying (which, in turn, requires good weather). Figure 4 shows an example of the graph which is generated for solving the problem of going from $l_0$ (a fact) to $l_1$ (a goal).

Estimated costs for flaws (boxes) and resolvers (ovals) are on their upper right. Notice that, in the example, the flaw $\varphi_0$ can only be solved by resolver $\rho_0$ which is directly applied (solid lines represent what is in the current partial solution). Additionally, since $\varphi_0 = \varphi(\sigma_3)$, the $active$ value is assigned to $\sigma_3$. The first flaw to be solved is, hence, $\varphi_1$, which can be solved either with resolver $\rho_1$, having an estimated cost of 3, or with resolver $\rho_2$ having an estimated cost of 4[6]. Applying, for example, the least expensive resolver $\rho_1$ would lead, as a consequence of constraint propagation, to the activation of the flaw $\varphi_2$ (notice that $precs(\rho_1) = \{\varphi_2\}$ and $cause(\varphi_2) = \{\rho_1\}$) which can be solved with the sole resolver $\rho_3$, which in turn activates the flaw $\varphi_4$ which is

---

[6]In the figure, the estimated costs are represented in the upper right of the flaws/resolvers and are computed through the $h_{max}$ heuristic. Whenever they do not coincide, in parenthesis is also represented the value from the $h_{add}$ heuristic.

solved with resolver $\rho_5$ leading to a solution. Finally, since $\varphi_4 = \varphi(\sigma_6)$, the *unified* value is assigned to $\sigma_6$.

## Current results

The causal graph, described in the previous section, has been implemented within the ORATIO solver[7]. In order to show the effectiveness of the proposed approach, we tested the solver, enhanced with the above heuristic, on different instances of the GOAC domain. Specifically, the Goal Oriented Autonomous Controller (GOAC) was an ESA initiative aimed at defining a new generation of software autonomous controllers to support increasing levels of autonomy for robotic task achievement. In particular, the domain, initially defined in (Fratini et al. 2011) and more recently cited in (Coles et al. 2019), aims at controlling a rover to take a set of pictures, store them on board and dump the pictures when a given communication channel was available. The interesting aspect of this domain is that communication can only take place within specific visibility windows that take into account the astronomical motions of the planets/satellites which, in some cases, may stand between the transmitting and receiving stations. The presence of these visibility windows, in particular, requires an explicit modeling of temporal aspects in order to adequately plan the transmission of information and can hence easily be modeled through, and solved by, timeline-based planners. The problem is made more interesting by the presence of constraints which include the available resources (e.g., memory and battery) as well as by having a distance matrix, among the possible locations, which might be not completely connected.

Figure 5 shows the execution times of different timeline-based solvers (i.e., EPSL (Cesta, Orlandini, and Umbrico 2013), AP2 (Fratini et al. 2011), J-TRE (De Benedictis and Cesta 2012), one of the precursors of ORATIO using a less accurate heuristic (De Benedictis and Cesta 2016), and the more recent PLATINUm (Umbrico et al. 2017)) as well as a couple of temporal-planning solvers (i.e., OPTIC (Benton, Coles, and Coles 2012) and COLIN (see (Coles et al. 2012)), both based on a classic FF-style forward chaining search (Hoffmann 2001)) in solving different instances of the GOAC problem. In particular, problems are obtained by varying the problem complexity along the number of pictures to be taken and the number of communication windows. Among all the generated problem instances, in particular, the ones with higher number of required pictures and higher number of visibility windows result as the hardest ones. The right mix of causal and temporal aspects makes the GOAC problem particularly complex to the point that some of the planners, beyond a certain number of pictures to collect and data dumps, show serious scalability issues. As shown in the figure, besides being considerably more efficient, compared to other timeline-based planners, ORATIO is also able to solve more complex instances. Compared to the temporal-planning solvers, however, it is clear that, despite significant improvements, there is still a performance gap to fill. Possible explanations of this gap in-

---

[7]https://github.com/pstlab/oRatio

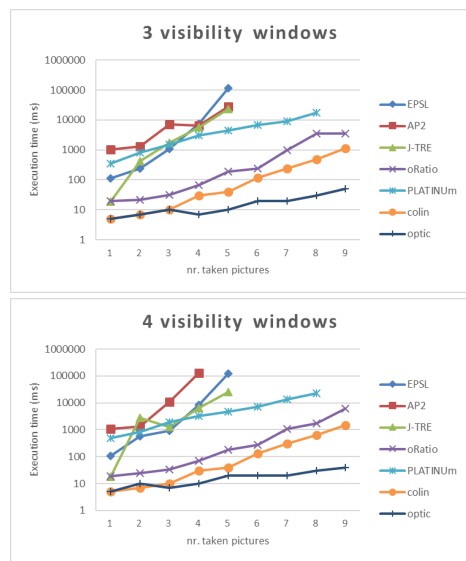

Figure 5: Execution times of different solvers to instances, of increasing complexity, of the GOAC problem.

clude the maintenance, in the current state of the solvers, of the consistency between the various constraints (which is not required in the forward state space search planners), in addition to the greater effectiveness of the FF heuristics. Another aspect to take into consideration regards the possibility of making the graph more accurate, so as to be able to represent heuristics as $h^2$ (Haslum and Geffner 2000; Haslum, Bonet, and Geffner 2005). Since it is not possible to recognize the mutual exclusivity between the resolvers directly from the rules' structure, we have not yet found an effective approach for implementing it.

**Comments on the results.** Although, for the moment, there are solvers able to solve the GOAC problem more efficiently than the ORATIO solver, we believe that the current results are nevertheless significant. In the first place, indeed, the heuristic described in this document proposes a complete paradigm shift for timeline-base planners: we pass from heuristics based on the current partial solution (i.e., the current token network $\pi$) to heuristics based on all possible plans that can be generated from starting from the planning problem. In so doing, we have the possibility to anticipate the consequences of decisions before they are even taken and this results in more accurate plan synthesis. A second aspect to consider regards the possibility of modeling (and, above all, integrating) different kinds of reasoning which depart from those more closely related to automated planning. By removing the temporal parameters from the tokens, indeed, we obtain a form of reasoning which is similar to constrained logic programming. The proposed heuristics, in particular, still remain valid, and paves the way for the efficient integration of different forms of reasoning such as, for example, automated planning and semantic reasoning. To better understand this aspect we can consider as an example the execution of Prolog program, whose efficiency strongly de-

pends on the order in which the goals are defined within the rules as well as on the order in which the rules are defined. Different rules having the same goals defined in a different order are, indeed, semantically equivalent. The programmer, however, could be wrong at defining such orders or, even worse, the most efficient order could depend on the value of the parameters, unavoidably affecting the performance of the resolution process. The introduction of heuristics such as those presented would alleviate these types of problems.

## Conclusions

The reasons for introducing a new timeline-based formalism are manifold and range from the possibility to model, through a uniform formalism, continuous changes over time (see, for example, Figure 1a) to make the plans more flexible in the execution phase (relaxing the constraint, present in some formalisms, that forces the timelines to be completely filled over time). Whatever the formalism, reasoning upon these systems remains particularly challenging from a computational point of view. For this reason we have introduced a new heuristic that takes into account, before starting the search, *all* possible resolvers for *all* possible flaws that may emerge from the resolution process, so as to be able to make choices according to a more accurate criterion. Although encouraging, the results show that there is still work to be done. As an example, since it is possible to recognize mutex resolvers by propagating constraints, it is worth to investigate different approaches for representing the $h^2$ heuristic. Analogously, the proper adaptation of landmark-based heuristics, might represent a fruitful path toward the resolution efficiency. We hence believe that, through this document, we can lay the foundations for the definition of a new typology of heuristics for the efficient resolution of timeline-based planning problems.

**Acknowledgments.** Authors work is partially supported by the INdAM-GNCS project *Metodi formali per tecniche di verifica combinata*, and by SI-ROBOTICS [8]. They are members of the OVERLAY[9] network.

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
