# OpenReview forum: "Investigating Lifted Heuristics for Timeline-based Planning"
_icaps-conference.org/ICAPS/2020/Workshop/HSDIP — HSDIP 2020_

### Official Review · AnonReviewer2 · 2020-04-01
**Initial assessment**

**Rating:** 6
**Confidence:** 3

**Review:**



The submission proposes the use of classical planning heuristics in the context of timeline-based planning.
As such, it clearly is within scope of the workshop topic, and could be considered for presentation.

This being just an informal assessment of the submission, I will just briefly outline three areas where
I think it could be strenghtened, which relate to
(1) formalization of problem, (2) the presented heuristic, and (3) the empirical results:


(1) The formalization of the timeline-based planning problem is announced as one of the paper contributions,
but it is not fully clear to me what exactly is part of previous research and where the actual contribution
starts, since Section 2 mixes an overview of related formalizations with the definition of the formalization
itself. If the formalization is novel and to be considered a part of the contribution, it should be much
clearer how it differs from previous related formalizations.


* The formalization could be improved. Although most of the Timeline-based planning section is quite
intuitive, there are a lot of concepts and definitions introduced. I would strongly suggest moving
the example in Section "The lifted heuristic formulation" much earlier, and using it as a running
example with which the whole formalism is illustrated. In terms of clarity, I found the following points
not too clear:

- Def. 1 introduces constraint networks, but the definition doesn't say too much beyond the idea that
they are made up of variables and constraints, without saying what these are. Constraint networks are
pretty standard (and hence a citation here should be in order), so I would suggest either to spell out
some complete standard definition, which is quite simple, or don't use any definition at all for them.

- The definition of "requirement" is a bit confusing. In Def. 5, it seems it is equated to
the body of a rule, which seems to be "either a slave (or target) token, a constraint among tokens [...]),
a conjunction of requirements or a (priced) disjunction of requirements. To begin with, the concept of
"slave" or "target" token has not been introduced, so it's not clear what that means.
The notion of "priced disjunction" is only made intuitive thanks to a footnote.
In Def. 6, however, a "requirement" is defined again, repeating almost the same definition in Def. 5, but
not quite, as here a requirement can also be "a (fact or goal) token".


- I'm also a bit confused with the ordering with which definitions are introduced. Def. 4 defines a Token network
as a set of tokens plus a constraint network defined over the variables in these tokens. The Rules are defined
in Def. 5, but it's not clear what relation they bear with Token networks: according to the definition, they bear
none, but then it seems that the body of rules "must also be present in the token network". What network exactly?
Also, if that is a requirement of the definition, it should be inside the definition, not as a comment.

- As a side comment, the relation between constraint networks / CSPs and the timeline-based problem
could be better fleshed out.


(2) As for the formulation of the lifted heuristic, I think the submission should formulate more clearly
and again, formally, what exactly is the technique being proposed. I get the overall idea about tackling temporal
from causal aspects of the problem separately, and tackling the latter through classical planning techniques,
but how this "translation" is exactly made is not clear at all to me. What exactly is the AND/OR graph being used?
How exactly does the proposed technique mix the classical-planning-like heuristics with the SAT/SMT-like search
procedures?


(3) For the empirical results, it's not clear to me in what language are the instances of GOAC benchmarks used,
given that different types of temporal planning engines, some PDDL-like, are used.
Are you using different encodings of the same problem, or are all planners capable of dealing with the same
modeling language?

---

### Official Review · AnonReviewer1 · 2020-08-17

**Rating:** 8
**Confidence:** 3

**Review:**

This is an interesting contribution to the workshop, and to the field
of AI planning, however the paper is unclear on many points and I think
it could be improved. Some specific points below.

Page 3: In the example rule, At(?x) @ ..., why are the tokens in the
rule body subscripted with "g" (which I assume stands for "goal")?
If we are to interpret the rule as causal, it would seem more logical
that they are required to be facts, i.e., that At(?x) will be true
(over the interval [s,e], on timeline tau) if, and only if, preceded
by a driveto or flyto token ending at s, with destination ?x (as
suggested by the description in the next paragraph). The head token
does not have either a goal or fact annotation; should we view the
rule as applicable to either, with the tokens in the body to have
a matching class?

Definition 3: Can there be more than one rule with the same head
token? If so, how are they interpreted? Are they equivalent to having
a single rule with a disjunctive body?

Definition 8 is vague: What does it mean for a token to be "recognized
as semantically equivalent"? Muscettola had a notion of "collapsing"
tokens, which implies posting an equality between their parameters and
time point variables; a goal token is then "achieved" if it can be
collapsed with a fact token. (I'm not sure about the terminology that
he used.)

What does it mean that a rule is "applied"? Should we interpret the
rule as a logical constraint (as in ASP), i.e., the rule is "applied"
iff for every token matching the head there exists a mapping of tokens
and values satisfying the body? (and in that case, what does this mean
if there are multiple rules with the same head?)

Or is rule application a form of regression? The presence of a token in
the network is "justified" (I think that was Muscettola's term) if it
equals the head of a rule whose body is also justified?

The definition and discussion of the extraction function and difference
between the token network and timelines is somehwat of a distraction. I
understand that the authors wanted to make a clear separation. However,
the statement of the problem (Def. 6) and the solution (Def. 8) are
entirely in terms of the token network. In other words, one does not
need to know the extraction function(s) in order to state a timeline
planning problem, to solve it, or to verify a given solution.

Footnote 3: I'm not convinced that it is "easy" to construct a proof of
undecidability this way. However, I do agree that without further
restrictions on the types of timelines, the problem is likely to be
undecidable (for example, if timeline domains are unbounded numbers, it
should be possible to encode a counter-machine halting problem).

The description of how the heuristic is computed (section "The lifted
heuristic formulation") would benefit from some clarification. The
heuristic calculation (equations defining G(.)) is straightforward,
but the construction of the graph over which this is calculated is not.
The example graph in Figure 4 is quite helpful, but without having all
the rules that define this example domain, it still does not allow me
to work out the procedure that computed the graph.

However, the example graph does raise a question: When the predicate
At(?x) appears recursively in the derivation of At(?x), the rule nodes
have are given an empty disjunction (and a heuristic value of +inf),
I assume as a way to avoid cyclic justifications, like in h^max/h^add.
However, how would this work if the domain requires moving through
multiple locations (e.g., from l_0 to l_2 to l_1), as described in the
rovers domain in the experiments section? In that case, the driveto
token would have an additional "precondition" (conjunct in its rule)
stating that the location driven from is connected to the target
location, but I assume it would still require the agent to be "At"
a departing location. As shown in Figure 4, the only driveto action
with a finite heuristic value will be from the initial location.

In a problem where actions move along a "roadmap" (such as driveto),
will the heuristic detect the number of steps in the map needed to
reach the goal node in this map? (as, e.g., h^max will do). If so,
how?

In the experiments, the authors compare the timeline-based planner
enhanced with the new heuristic against three other timeline-based
planner, but not against FAPE or Europa, which are both well-known
examples of general-purpose timeline-based planners, and both
available in open source (https://github.com/laas/fape.git and
http://europa-pso.googlecode.com/svn/PLASMA/trunk). What is the
reason that these two planners are not considered?

---

### Comment · Program_Chairs · 2020-09-14
**Final Decision: Accept**

Dear Authors,

Thank you very much for your submission. We are happy to inform you that we have decided to accept it and we look forward to your talk in the workshop. You will receive additional information per mail in the coming days.

Best,
The HSDIP'20 team

---

### Decision · Program_Chairs · 2020-09-30

Accept